# Antioxidant and Photoprotective Activities of *Viola philippica* Polyol Extracts

**DOI:** 10.3390/antiox14070884

**Published:** 2025-07-18

**Authors:** Jiang Li, Jiancheng Ma, Ya Li, Lan Luo, Wenhuan Zhang, Yong Tian, Yuncai Tian, Yi Li, Zhongjuan Wang, Mingyi Wu

**Affiliations:** 1State Key Laboratory of Phytochemistry and Natural Medicines, Kunming Institute of Botany, Chinese Academy of Sciences, Kunming 650201, China; lijiang@mail.kib.ac.cn (J.L.);; 2University of Chinese Academy of Sciences, Beijing 100049, China; 3School of Pharmacy, Dali University, Dali 671003, China; 4Shanghai Zhenchen Cosmetics Co., Ltd., Shanghai 201415, China; 5Shanghai Zhizhenzhichen Technology Co., Ltd., Shanghai 201109, China

**Keywords:** *Viola philippica*, antioxidant, reactive oxygen species, polyol extracts, UVB

## Abstract

*Viola philippica* (VP), a traditional Chinese medicinal herb widely used for its antibacterial and antioxidant properties, has recently garnered attention for its potential in skin photoprotection. VP was extracted using glycerol (GLY), 1,3-propanediol (PDO), and 1,3-butanediol (BDO) at concentrations of 30%, 60%, and 90% (*w*/*w*) to evaluate its antioxidant and UV-protective properties. The total phenolic content (TPC) and total flavonoid content (TFC) of the nine extracts ranged from 34.73 to 71.45 mg GAEs/g and from 26.68 to 46.68 mg REs/g, respectively, with the highest TPC observed in 90% PDO and the highest TFC in 60% GLY. Antioxidant assays revealed IC_50_ values of 0.49–1.26 mg/mL (DPPH), 0.10–0.19 mg/mL (ABTS), and 1.58–460.95 mg/mL (OH). Notably, the 60% GLY, 30% PDO, and 90% PDO extracts demonstrated notable protective effects against UVB-induced cell damage, reducing intracellular ROS levels and preventing DNA damage. RNA-seq analysis revealed that the protective effects were associated with the modulation of key molecular pathways, including neutrophil extracellular trap formation and TNF, IL-17, and HIF-1 signaling. These findings suggest that *Viola philippica* polyol extracts, particularly those using 60% GLY, 30% PDO, and 90% PDO, have promising potential for skin photoprotection and could be utilized as natural antioxidants in cosmetic formulations.

## 1. Introduction

The skin, recognized as the largest organ in the human body, serves as the primary barrier against a variety of external threats and damage [1,2]. Its structure consists of three distinct layers, each with specific functions and characteristics: the epidermis, dermis, and subcutaneous tissue [3,4]. The outermost layer of the skin, known as the epidermis, is primarily composed of keratinocytes, melanocytes, and Langerhans cells. The epidermis plays a crucial role in protecting the body from external factors [5]. Situated beneath the epidermis, the dermis is crucial for both structural integrity and functional processes. It is rich in an extracellular matrix produced by fibroblasts and contains blood vessels, nerve fibers, and sweat glands [6]. The innermost skin layer, referred to as subcutaneous tissue, consists mostly of fat and plays a significant role in temperature control and absorbing shock [7]. When epidermal keratinocytes and dermal fibroblasts are subjected to ultraviolet (UV) radiation, it triggers cellular senescence, which contributes to the photoaging of the skin [8]. UV radiation is categorized into three types based on its wavelength: UVA (ranging from 320 to 400 nm), UVB (from 280 to 320 nm), and UVC (spanning 100 to 280 nm) [9]. Although UVC is the most destructive, it is completely absorbed in the ozone layer [10]. UVB accounts for 5% of ground sunlight ultraviolet radiation before reaching the Earth’s surface, and UVA accounts for 95% [11]. UVB can only reach the basal layer of the skin at most, while UVA can penetrate the skin’s epidermis and reach the dermis [12]. Although UVB accounts for a smaller proportion of ground sunlight ultraviolet radiation and can only reach the basal layer of the skin, it has stronger biological activity on the skin than UVA [13]. The prolonged exposure of the skin to UVB radiation results in the excessive accumulation of reactive oxygen species (ROS), which leads to DNA damage, inflammation, and the dysfunction of the skin barrier. These factors ultimately contribute to skin aging and the development of cancer [14]. Although UV radiation can cause health effects on the skin, various strategies have been explored to mitigate its harmful effects. Among these, plant extracts have attracted attention for their potential photoprotective properties [15,16,17]. These natural compounds, rich in antioxidants and anti-inflammatory agents, offer a promising approach to reducing UV-induced skin damage.

*Viola philippica* (VP), a violet genus of Violaceae, is recognized for its ability to clear heat, detoxify, cool the blood, and reduce swelling [18]. VP contains various active ingredients, such as anthocyanins, flavonoids, polysaccharides, and coumarins, which contribute to its pharmacological activities, including anti-inflammatory, antioxidant, analgesic, and antibacterial effects [19,20]. Esculetin is one of the active components in VP with DPPH and OH radical scavenging activity, and it protects cells from H_2_O_2_-induced lipid peroxidation and DNA damage by promoting the nuclear accumulation of Nrf2 [21]. Schaftoside exhibits potent anti-inflammatory and antioxidant activities, and it inhibits cellular ferroptosis by activating the Nrf2/GPX4 pathway [22]. Rutin, as a natural flavonoid compound, protects fibroblasts from UVA radiation [23]. Several studies have shown that VP extracts have anti-aging, antioxidant, and sunscreen properties [24,25,26]. In addition, esculetin and schaftoside in VP extract can activate CD44 and AQP3 in HaCaT cells, enhancing skin barrier function and hydration [27]. Given these benefits, the development of cosmetics containing VP extracts holds great promise.

However, it is worth noting that green, natural, and non-toxic extract raw materials are crucial for application in cosmetics. In the process of extracting and purifying the active ingredients of VP, organic solvents such as methanol, ethanol, and ethyl acetate are typically employed to efficiently extract the active components from the plant [28]. Despite this, there are certain limitations and risks associated with the application of these plant extracts prepared with organic solvents in cosmetics. Methanol has an irritating effect on human skin, the respiratory tract, and the conjunctiva, and long-term exposure may cause damage to the skin system and liver and kidney functions [29]. Alcohol can cause a loss of skin moisture, leading to dryness and roughness of the skin. It can also damage the skin barrier, and prolonged use may harm the stratum corneum, reducing the skin’s resistance [30]. Polyhydric alcohol solvents with different polarities (such as glycerol containing three hydroxyl groups and 1,3-butanediol with medium polarity) can optimize the extraction efficiency of target compounds (such as phenolic acids and flavonoids) through selective dissolution, ensuring the efficient retention of bioactive components. Additionally, polyhydric alcohols like glycerol and 1,3-butanediol are commonly used as base ingredients in cosmetics, being low in toxicity and non-irritating to the skin, which allows the extracts to be used directly in cosmetic products [31,32]. Several studies have utilized polyol solvents to extract bioactive compounds from plants such as *Chlorella vulgaris* [33] and Silybum marianum [34], camellia seed dregs [35], and *Rhus chinensis* leaves [36]. Despite the clear advantages of polyols as extraction solvents, their application in the preparation of natural and effective cosmetic ingredients remains relatively underutilized.

Consequently, in this study, three polyols (glycerol, 1,3-propanediol, and 1,3-butanediol) with lower toxicity and better biocompatibility were used as solvents to extract *Viola philippica*. The bioactive compounds, in vitro antioxidant activities, and photoprotective potentials of nine VP extracts were evaluated. In addition, the potential molecular mechanisms involved in the antioxidant effects of VP extracts on HaCaT cells exposed to UVB were also explored in this study.

## 2. Materials and Methods

### 2.1. Chemicals and Reagents

All chemicals and reagents used in this study were of analytical grade. Esculin, esculetin, luteolin, naringenin, rutin, caffeoyltartaric acid, chicoric acid, and schaftoside were purchased from BioBioPha (Kunming, China). Glycerol (GLY), 1,3-propanediol (PDO), and 1,3-butanediol (BDO) were acquired from Shanghai Hanhong Technology Co., Ltd. (Shanghai, China). 2,2′-Azino-bis-3-ethylbenzothiazoline-6-sulfonic acid (ABTS) and 1,1-diphenyl-2-picrylhydrazyl radical (DPPH) were obtained from Shanghai Myriad Chemical Technology Co., Ltd. (Shanghai, China). Methanol and acetonitrile for HPLC (purity ≥ 99.9%) were from Merck (Darmstadt, Germany). Dulbecco’s modified eagle medium (DMEM) and penicillin/streptomycin mixture were from Gibco (New York, NY, USA). Fetal bovine serum (FBS) was purchased from Lonsera (Suzhou, China). Comet assay kit was from Beyotime (Shanghai, China). 2′,7′-Dichlorofluorescein diacetate (DCFH-DA) fluorescent probe was from Dalian Meilun Biotechnology Co., Ltd. (Dalian, China).

### 2.2. Preparation of Polyol Extracts from Viola philippica

The dried whole plant of *Viola philippica* (VP) was purchased in Bozhou, Anhui Province, and identified by Prof. Wu Mingyi of the Kunming Institute of Botany, Chinese Academy of Sciences. The procurement and identification of the plant species were conducted in strict accordance with the 2020 Edition of the Pharmacopoeia of the People’s Republic of China, which specifies that *Viola philippica* should be harvested and sun-dried during the spring and autumn seasons annually. The dried material was ground into a fine powder using a laboratory mill. A total of 10 g of the whole-herb powder of VP was mixed with 100 mL of solvent systems containing glycerol (GLY), 1,3-propanediol (PDO), and 1,3-butanediol (BDO) at concentrations of 30%, 60%, and 90% (*w*/*w*). The extraction was performed at a constant temperature of 60 °C for 2 h. Additionally, extraction was conducted using 60% ethanol (EtOH) (*w*/*w*) under the same conditions. The extraction process was carried out using a simple maceration method, which is suitable for polyol solvents, as the high viscosity and low volatility of polyols allow them to maintain a stable extraction efficiency over a longer period of time. After completing the extraction process, the mixed solution was filtered using a 0.45 μm membrane, and the filtrate was collected to obtain extract solutions from various solvent systems. All extraction solutions were stored at 4 °C in the dark. To prevent the oxidation and degradation of polyphenols in the extracts, nitrogen (N_2_) was used to displace the gas in the containers, preparing for subsequent activity assays. It was observed that under these conditions, the extraction solutions could be stably preserved for at least six months.

### 2.3. Total Phenolic and Flavonoid Content Determination

The total phenolic content of the samples was assessed through the Folin–Ciocalteu method, utilizing gallic acid as a reference standard [37]. In brief, 1 mL of a sample solution (2 mg/mL) was combined with 5 mL of Folin–Ciocalteu reagent. After a 5 min interval, 4 mL of 7.5% Na_2_CO_3_ was incorporated into the mixture, which was then allowed to react at room temperature for 1 h. The absorbance was then measured at 765 nm. A standard curve was established using gallic acid concentrations ranging from 20 to 120 μg/mL on the x-axis and absorbance on the y-axis (equation: y = 97.03x + 0.032, R^2^ = 0.9978). The total polyphenol content in the extract was calculated based on this standard curve and expressed in mg gallic acid equivalents (GAEs)/g. Each experiment was carried out in triplicate, and the results are expressed as mean values ± standard deviations.

The total flavonoid content in the VP extracts was determined, with rutin as the standard [38]. To 2 mL of the extracts (10 mg/mL), 0.5 mL of sodium nitrite solution was added, after which the mixture was thoroughly shaken and left to stand for 6 min. Next, the mixture was shaken again, and 0.5 mL of aluminum nitrate solution was added, followed by another round of shaking. After resting for an additional 6 min, 4 mL of sodium hydroxide solution was included. The solution was mixed thoroughly, and the absorbance was measured at 510 nm. A standard curve was constructed with the rutin mass concentration plotted on the *x*-axis and absorbance values on the *y*-axis. The regression equation obtained was y = 12.71x − 0.0049, R^2^ = 0.9996. The results are expressed as rutin equivalents per gram of dried *Viola philippica* (mg REs/g). All experiments were conducted in triplicate.

### 2.4. Analysis of Active Components in VP Extracts by HPLC and LC-MS

The active compounds within the VP extract were identified utilizing the Agilent HPLC 1100 series equipped with an Agilent ZORBAX SB-C18 column (250 mm × 4.6 mm, 5 µm, Santa Clara, CA, USA). The mobile phase comprised acetonitrile and a 0.1% phosphoric acid solution, following a gradient elution program: at 0 to 25 min, 8% acetonitrile and 92% phosphoric acid were used, and from 25 to 40 min, acetonitrile increased from 8% to 60%, while the phosphoric acid solution decreased from 92% to 40%. The wavelength for detection was established at 344 nm, while the column temperature remained at 30 °C, with an injection volume of 10 µL and a flow rate of 0.1 mL/min. Standard reference compounds included esculin, caftaric acid, esculetin, caffeic acid, chlorogenic acid, schaftoside, rutin, chicoric acid, luteolin, and naringenin; these were prepared by dissolving each one in MeOH at a concentration of 0.1 mg/mL. To create calibration curves, the standard solutions were sequentially diluted to achieve the necessary concentrations. Subsequently, these solutions were filtered through a 0.22 µm membrane filter, and the VP extract, which was diluted fivefold, was quantified using the established calibration curves. Calibration functions for the compounds were determined based on the peak area (Y), concentration (X, µg/10 µL), and mean ± standard deviation (*n* = 3).

LC-MS analysis was performed to confirm the identity of the compounds. The ESI source was operated in both positive and negative ionization modes to comprehensively detect the compounds. The mass spectrometer was set to scan a mass-to-charge ratio (*m*/*z*) range of 20–2000. The collision energy was set to 10 V, and the fragmentor voltage was 135 V. The data acquisition was performed in full-scan mode, and the compounds were identified based on their molecular ion peaks and fragmentation patterns. The identified compounds included esculin, esculetin, schaftoside, rutin, and other bioactive components.

### 2.5. Determination of Antioxidant Potential

#### 2.5.1. DPPH Free Radical Scavenging

DPPH free radical scavenging was assessed on the basis of the method used by Li et al. [39], with minor modifications, as elaborated below. The concentration of the VP extract stock solution was 100 mg/mL, which was diluted 5, 25, 125, 625, and 3125 times with ultrapure water to prepare the test solution. In a 96-well plate, 100 μL of the test solution and 100 μL of DPPH working solution were added to each well, and the reaction was carried out for 30 min protected from light, and the absorbance value was measured at 540 nm. In total, 100 μL of the test solution and 100 μL of anhydrous ethanol were used in sample control wells; 100 μL of DPPH working solution and 100 μL of anhydrous ethanol were used in blank control wells.

#### 2.5.2. ABTS Free Radical Scavenging

ABTS free radical scavenging was assessed on the basis of the method used by Li et al. [40] with minor modifications. In total, 4.06 mg/mL ABTS solution was mixed with 0.593 mg/mL ammonium persulfate solution at a volume ratio of 1:1, reacted at 4 °C for 12 h, and then diluted 30-fold to obtain the ABTS working solution before use. In a 96-well plate, 20 μL of the test solution was added, and 180 μL of ABTS working solution was added to the sample group, and the absorbance value was measured at 734 nm after 30 min of reaction protected from light; 20 μL of the test solution and 180 μL of the working solution in ultrapure water were used in the sample control wells; and 20 μL of ultrapure water and 180 μL of the ABTS working solution were used in the blank control wells.

#### 2.5.3. OH Scavenging

The ability of VP extracts to inhibit OH· activity was determined using the Fenton method [41], with some modifications. In a 96-well plate, 40 μL of 3.6 mM salicylic acid solution, 0.88 mM hydrochloric acid solution, 2.5 mM ferrous sulfate solution, test solution, and hydrogen peroxide solution were added sequentially; the reaction was carried out at 37 °C for 20 min after mixing, and the absorbance value was measured at 492 nm. For the sample control well, 40 μL of the test sample solution mixed with 160 μL of ultrapure water was used. For the blank control well, 40 μL of salicylic acid solution, 40 μL of hydrochloric acid solution, 40 μL of ferrous sulfate solution, 40 μL of ultrapure water, and 40 μL of hydrogen peroxide solution were sequentially added.

The VP extract solution was diluted 50 times with ultrapure water in five iterations, and the free radical scavenging rate was calculated, from which the half-maximal inhibitory concentration IC_50_ value was determined. The DPPH, ABTS, and OH· free radical scavenging activity of the sample was calculated using the following formula: free radical scavenging rate (%) = [(Ac − (As − Ad))/Ac] × 100%. Ac is the absorbance value of the blank well; As is the absorbance value of the sample well; and Ad is the absorbance value of the sample control well.

### 2.6. Ultraviolet Wavelength Scanning of VP Extracts

An appropriate amount of extract stock solution was diluted 50 times with deionized water and mixed thoroughly, and then 1 mL of the extract dilution was taken into a quartz cup, and the UV absorption spectrum in the range of 200–400 nm was recorded by a UV spectrophotometer.

### 2.7. Cellular Analysis of VP Extracts

#### 2.7.1. Cell Culture and Treatment

HaCaT cells were purchased from the Cell Bank of the Chinese Academy of Sciences (Shanghai, China). The cells were maintained at 37 °C under 5% CO_2_ in DMEM supplemented with 10% fetal bovine serum, 100 U/mL penicillin, and 100 μg/mL streptomycin. For the photodamage induction model, HaCaT cells were treated with 1% VP extracts (diluted 100-fold with DMEM medium) for 2 h and subsequently exposed to UVB radiation (312 nm, 3 mW/cm^2^) for 15 min, delivering a total dose of 0.5 J/cm^2^ using a UVB irradiation device (Bio-SUN, VILBER, Collégien, France). After irradiation, the cells were incubated for an additional 24 h before undergoing subsequent testing.

#### 2.7.2. Cell Viability

HaCaT cells were seeded at a concentration of 1 × 10^4^ cells per well in a 96-well plate and incubated in a cell culture incubator set at 37 °C with 5% CO_2_ for a duration of 24 h. After this period, the cells received treatment with 1% *Viola philippica* polyol extracts for 24 h. Subsequently, 10 μL of CCK-8 solution was introduced to each well, and the plate was kept in the dark at 37 °C for 2 h. After this incubation, the optical density (OD) was recorded at 450 nm using a microplate reader. LDH quantitatively released into the cell culture supernatant after 24 h of VP extract treatment was detected using an LDH kit (Beyotime Biotechnology, Shanghai, China) and expressed as a percentage of total cellular LDH (100% lysis).

#### 2.7.3. VP ROS Assay

HaCaT cells (1 × 10^5^ cells) were seeded into a 24-well plate and cultured for 24 h. The cells were treated with 1% VP extracts and subsequently exposed to UVB irradiation. After 24 h, the cells were washed with PBS, followed by the addition of 500 μL DCFH-DA solution (dissolved in DMSO; final concentration of 10 μM), and incubated at 37 °C for 30 min. The incubation solution of DCFH-DA was removed, and the cells were washed three times with serum-free culture medium to remove unabsorbed DCFH-DA. After washing, 500 μL of PBS solution was added to each well. The fluorescence of ROS was measured using a flow cytometer (BD FACSCelesta 3, Franklin Lakes, NJ, USA) at an excitation/emission wavelength of 485/525 nm.

#### 2.7.4. Comet Assay

HaCaT cells were cultured in 12-well plates at a density of 1 × 10^5^ cells per well and allowed to incubate overnight to ensure proper adhesion. After treating the cells with 1% VP extract for 2 h, they were exposed to UVB irradiation at 0.5 J/cm^2^. After 24 h incubation, the cells were collected, washed with PBS, and then resuspended in 150 μL of PBS. The neutral comet assay was then conducted using the comet assay kit in accordance with the manufacturer’s guidelines, ensuring that the procedure was repeated three times for the consistency and reliability of the results. Cell images were captured using a confocal microscope (TCS SP8, Leica, Wetzlar, Germany) to observe the extent of DNA damage and calculate the tail moment length.

### 2.8. RNA-Seq

After UVB irradiation, total RNA was carefully extracted from HaCaT cells utilizing Trizol reagent according to the manufacturer’s guidelines. The quantity and quality of the RNA obtained from HaCaT cells were evaluated using the Agilent Technologies Bioanalyzer 2100 system (Santa Clara, CA, USA). Subsequently, cDNA library construction was performed alongside quality inspection, clustering, and sequencing, all carried out in accordance with the established protocols provided by Novogene (Beijing, China). In the analysis of the data, differentially expressed genes (DEGs) were identified, with specific criteria set to classify them as significantly differentially expressed. Genes with a |log_2_FC| > 1 and a Padj value < 0.05 were highlighted as significant. Furthermore, an in-depth annotation of the pathways and functional relevance of these DEGs was conducted, utilizing enrichment analysis through the Kyoto Encyclopedia of Genes and Genomes (KEGG) and Gene Ontology (GO).

### 2.9. Data Analysis

Data were analyzed statistically and graphed utilizing GraphPad Prism 8.0. The results are displayed as the mean ± standard deviation (SD), and the significance of the experimental findings was evaluated using one-way ANOVA, complemented by Tukey’s multiple comparisons test for post hoc analysis. *p* < 0.05 is considered statistically significant.

## 3. Results

### 3.1. Analysis of Total Phenolic and Total Flavonoid Content of VP Polyol Extracts

Phenolic and flavonoid compounds are known for their strong antioxidant properties. As shown in Table 1, there were significant differences in the content of active compounds obtained from the extraction of different concentrations and types of polyols. In terms of the phenolic content, the total phenolic content (TPC) of the 60% EtOH extract was measured at 63.58 ± 0.05 mg gallic acid equivalents (GAEs)/g. The TPCs of the 60% BDO, 60% PDO, and 90% PDO extracts were significantly higher than that of the 60% EtOH extract (*p* < 0.001). Conversely, the TPCs of the 30% GLY, 90% GLY, and 90% BDO extracts were significantly lower than that of the 60% EtOH extract (*p* < 0.001). Regarding the total flavonoid content (TFC), the TFC of the 60% EtOH extract was 43.33 ± 0.07. All extracts, except for the 60% GLY extract, exhibited lower TFCs compared to the 60% EtOH extract, with the 60% GLY extract demonstrating a significantly higher TFC (46.68 ± 0.27).

### 3.2. Quantitative Analysis of Active Compounds in VP Extracts via HPLC

The phytochemical composition of VP extracts was investigated with an emphasis on their potential as a source of bioactive compounds. High-performance liquid chromatography (HPLC) was employed to identify and quantify the active compounds in the VP extracts. As shown in Figure 1 and Appendix A, the active components in the VP polyol extracts are consistent with those in the 60% ethanol extract, both containing significant amounts of esculin, esculetin, schaftoside, and rutin. Esculetin is considered to be the characteristic active ingredient in VP, and the HPLC results showed that all the extracts contained more than 200 μg/mL of esculetin, except the GLY (90%) extract which contained 143.81 μg/mL of esculetin, with the highest contents in PDO (90%) and BDO (60%). Although the chromatographic peaks of esculin were not exactly the same as those of the standard, it was confirmed to be esculin by LC-MS (Appendix A).

### 3.3. Free Radical Scavenging Ability of VP Extracts

The DPPH radical, ABST radical, and OH· radical are commonly used to evaluate the antioxidant capacity of extracts (Table 2). The results of the IC_50_ measurements of half of the concentrations of scavenged DPPH radicals showed that the DPPH radical scavenging ability of several VP extracts differed: 60% GLY had the lowest IC_50_ value of 0.49 mg/mL and stronger antioxidant activity, and 90% GLY had the highest IC_50_ value and weaker antioxidant activity. For ABTS radical scavenging activity, the VP extracts had IC_50_ values ranging from 0.1 to 0.19 mg/mL with strong antioxidant activity; for OH· radical scavenging activity, 30% PDO had the lowest IC_50_ of 1.58 mg/mL, and 90% PDO and 90% BDO had weaker activities, with an IC_50_ of 339.45 and 460.95 mg/mL, respectively.

### 3.4. Effect of Ultraviolet Wavelength Absorption by Extracts of VP

The sun protection performance was evaluated by measuring the absorbance of VP extracts in the range of 200–400 nm. As shown in Figure 2, all extracts exhibited good UV absorption across the entire spectrum, but their intensities varied. The 90% GLY extract demonstrated strong UV absorption, followed by 60% BDO and 90% PDO, while 30% GLY showed relatively weaker UV absorption.

### 3.5. Protection Against UVB-Induced HaCaT Cell Damage by VP Extracts

#### 3.5.1. Effect of VP Polyol Extracts on the Activity of UVB-Induced HaCaT Cells

The cytotoxicity of polyol alcohol and VP polyol extracts on HaCaT cells and their effects on the survival rate of HaCaT cells after UVB irradiation were evaluated through CCK-8 experiments. The polyol itself, when diluted 100 times with the culture medium, showed no significant effect on HaCaT cells (Appendix A). The VP polyol extract at a concentration of 1% showed no significant toxicity to HaCaT cells, and all extracts significantly promoted the proliferation of HaCaT cells, except for GLY (90%) and BDO (30%) (Figure 3A). The LDH release assay further confirmed the non-cytotoxicity of the VP extract (Appendix A). In subsequent experiments, HaCaT cells were pretreated with 1% VP extracts for 2 h and subsequently exposed to 0.5 J/cm^2^ UVB. The experimental results showed that all the extracts significantly increased the UVB-induced decrease in HaCaT cell viability, except for BDO (90%), which was able to significantly reduce the UVB-induced decrease in HaCaT cell viability (Figure 3B).

#### 3.5.2. Effect of VP Polyol Extracts on UVB-Induced Intracellular ROS in HaCaT Cells

UVB irradiation leads to the disruption of the balance between intracellular ROS production and the antioxidant defense system, resulting in oxidative stress. The ROS scavenging ability of VP extracts was evaluated by measuring the fluorescence intensity of DCF-DA using flow cytometry. As shown in Figure 4, the intracellular ROS level significantly increased after UVB irradiation at 0.5 J/cm^2^; except for PDO (90%), all other extracts significantly reduced the ROS level in HaCaT cells.

#### 3.5.3. Effect of VP Polyol Extracts on UVB-Induced DNA Damage in HaCaT Cells

The comet assay is commonly used to assess DNA damage. When HaCaT cells were treated with VP extract without any exposure to UVB radiation, the results indicated that there was no noteworthy effect on DNA damage (Appendix A). As shown in Figure 5, after the UVB irradiation of HaCaT cells, the average cell trailing length of the model group was 73.30 μm, while that of the blank group was 0.53 μm, indicating that UVB can cause DNA damage in HaCaT cells. The average trailing lengths of HaCaT cells were significantly reduced by the GLY (60%), PDO (30%), and BDO (90%) extracts, with average trailing lengths of 51.20 μm, 58.18 μm, and 60.35 μm, respectively, suggesting that these three extracts have a protective effect on UVB-irradiated HaCaT cells.

### 3.6. RNA-Seq Analysis

To elucidate the underlying molecular mechanisms and provide evidence to confirm the protective effects of VP extracts, RNA-seq was performed to screen for key molecular pathways. Differential genes (DEGs) were screened by |log_2_FC| > 1 and Padj < 0.05, and volcano maps were drawn. Red dots indicate significantly up-regulated genes, while blue dots indicate significantly down-regulated genes. As shown in Figure 6A, a total of 516 DEGs were identified after UVB treatment, in which 262 genes were up-regulated and 254 genes were down-regulated. When treated with 30% PDO extract, 372 DEGs were identified, of which 108 genes were up-regulated and 264 genes were down-regulated (Figure 6B). Differential genes were selected to generate cluster heatmaps, with each row representing one gene (Figure 6C). GO enrichment analysis of the DEGs showed that the protective effect of PDO (30%) against UVB-induced HaCaT cell damage mainly involved biological processes such as the cellular response to stimulus, signaling, the oxidation–reduction process (Figure 6D). KEGG pathway enrichment analysis showed that PDO (30%) attenuated UVB-induced photodamage in HaCaT cells mainly through the formation of neutrophil extracellular traps, IL-17 signaling pathway, HIF-1 signaling pathway, and TNF signaling pathway (Figure 6E).

## 4. Discussion

The present study investigated the antioxidant activities and photoprotective potential of nine *Viola philippica* (VP) extracts prepared using different polyol solvents. Our research findings indicate that extracts based on glycerol, 1,3-propanediol, and 1,3-butanediol effectively retain bioactive compounds while demonstrating significant antioxidant and UV protection effects, highlighting their advantages over traditional organic solvents.

Polyols have emerged as green solvents for plant bioactive compound extraction, offering distinct advantages over conventional organic solvents. As non-toxic, hygroscopic substances, polyols can preserve the integrity of compounds while allowing for direct use in cosmetic applications after filtration [42]. Their selective solubility characteristics for target molecules, such as phenols and flavonoids, result in extraction efficiencies that surpass those of conventional solvents like methanol and ethanol, all while maintaining excellent safety profiles for both skin and the environment [35]. The glycerol extracts of *S. marianum* contain a similar amount of silymarin to ethanol extracts while also exhibiting notable antioxidant, anti-elastase, anti-tyrosinase, and anti-inflammatory activities, rendering them suitable for the formulation of high-value anti-aging products [34]. The extracts derived from Camellia seed dregs using polyol demonstrate significantly enhanced DPPH scavenging efficiency and total polyphenol and flavonoid content in comparison to those using traditional solvents such as methanol and ethanol [43]. Additionally, the polyol extracts of *Canthium horridum* Blume leaves display markedly superior antioxidant and anti-tyrosinase activities compared to ethanol extracts [44]. Water–alcohol extracts of VP have been demonstrated to possess antioxidant potential [25]. Therefore, this study employed greener and safer polyols for the extraction of VP and evaluated these extracts’ active components and bioactivity. The experimental results indicated that the extracts obtained using polyols exhibited total phenols, total flavonoids, and free radical scavenging activity that were superior to or comparable with those of traditional ethanol extracts. This suggests that polyol solvents, as emerging extraction solvents, have significant application prospects in the fields of cosmetics and pharmaceuticals.

The superior performance of polyol extracts is further exemplified in our experimental results, where specific extracts showed exceptional phenolic and flavonoid content. The total phenolic content in the 60% BDO, 60% PDO, and 90% PDO extracts was significantly higher than that in EtOH (60%), and GLY (60%) had a higher total flavonoid content than EtOH (60%). This indicates that solvents have selective effects on extract composition. GLY is effective in extracting flavonoids, while BDO and PDO are better for phenolics. The three hydroxyl groups of glycerol may preferentially solubilize polar flavonoids such as rutin, whereas the moderate polarity of 1,3-butanediol (logP = −0.24) improves the extraction of less hydrophilic phenolic acids [31,32]. The solvent concentration also matters, with 30–60% polyols generally being optimal for extracting phenolics and flavonoids [45,46]. The highest total phenolic content was found in BDO (60%), and the highest total flavonoid content was in GLY (60%). This may be because the 60% solvent concentration achieves the best balance between solubility and permeability, thus improving the extraction efficiency. The higher phenolic and flavonoid content of the polyol extract may have contributed to its enhanced free radical scavenging ability. The free radical scavenging activity of the PDO (90%) and BDO (90%) extracts is relatively weak, which may be due to the reduced solubility of key antioxidant compounds at high solvent concentrations [46]. The GLY (60%) extract exhibited superior DPPH/ABTS scavenging ability, and this enhanced activity was associated with its high flavonoid content, as rutin and esculetin demonstrate synergistic radical neutralization through hydrogen donation and electron transfer mechanisms [47]. The GLY (90%) extract demonstrated the strongest absorption capacity in the UV absorption experiment but exhibited the weakest DPPH scavenging ability. This suggests that the photoprotective effect of this extract may primarily rely on its UV absorption capacity rather than its free radical scavenging ability. The GLY (90%) extract may be rich in UV-absorbing compounds (such as coumarins) but lack flavonoid compounds that efficiently scavenge free radicals (such as rutin). This finding indicates that when evaluating the photoprotective potential of plant extracts, both their UV absorption capacity and antioxidant activity should be considered simultaneously. Interestingly, the PDO (30%) extract showed enhanced OH^·^ scavenging capacity, suggesting that its polarity is optimal for the extraction of specific Fe^2+^-chelating compounds [48].

HPLC analysis revealed that the VP extract was rich in various known antioxidant substances, such as esculin, esculetin, schaftoside, and rutin. These compounds synergistically scavenge free radicals through hydrogen donation and electron transfer mechanisms, thereby enhancing the antioxidant capacity of the extract. Esculetin, a coumarin compound, can protect cells from oxidative damage by promoting Nrf2 nuclear accumulation [21]. Additionally, rutin, a natural flavonoid, can protect fibroblasts from UVA radiation by activating the Nrf2 pathway [23]. The presence of these known antioxidant substances provides a molecular basis for the photoprotective potential of VP extracts.

UVB irradiation can cause DNA damage and oxidative stress in skin cells, which is one of the main causes of skin photoaging [49,50]. Given the strong free radical scavenging ability and antioxidant capacity of VP extracts, a UVB-induced HaCaT cell model was further analyzed to evaluate the photoprotective effects and potential mechanisms of VP extracts. The results showed that these extracts significantly protected HaCaT cells against UVB-induced damage, as indicated by the increased cell viability and reduced intracellular ROS levels. Additionally, the comet assay revealed that these extracts prevented UVB-induced DNA damage, further confirming their photoprotective effects. These findings suggest that the antioxidant properties of VP extracts contribute to their ability to protect skin cells from UV-induced damage, highlighting their potential application in photoprotection. Although antioxidant activity is an important mechanism of photoprotection, this study found that the 30% PDO extract exhibits strong ROS scavenging activity and DNA-protective effects, while its DPPH and ABTS antioxidant activities are not prominent. This suggests that the mechanism of photoprotection may not solely rely on antioxidant activity but may also involve other mechanisms.

RNA-seq analysis revealed that the PDO (30%) extract significantly alleviated UVB-induced cellular damage by modulating the IL-17, HIF-1, and TNF signaling pathways. These signaling pathways are closely associated with oxidative stress, DNA repair, and apoptosis, suggesting that photoprotection may be achieved through the synergistic action of multiple mechanisms. IL-17 promotes UVB-induced skin damage by activating a series of downstream NF-κB and MAPK signaling pathways through the activation of its receptor IL-17R [51]. HIF-1a is stably expressed in basal epidermal keratinocytes, acts as a sensor of environmental hypoxia, and regulates systemic metabolic responses [52]. In UVB-irradiated keratinocytes, ROS induced HIF-1a transcription and initiated apoptosis [53]. Furthermore, epidermal HIF-1a deficiency reduced photocarcinogenesis by enhancing DNA repair and ROS production [54]. These results suggest that VP extracts may exert photoprotective effects through the IL-17, HIF-1, and TNF signaling pathways. However, a more comprehensive exploration and analysis of the underlying molecular mechanisms is needed to fully understand the mechanism of action of VP extracts.

Given their strong free radical scavenging activity and protection against UVB-induced oxidative stress and DNA damage, VP extracts have potential use as cosmetic additives. However, there are some limitations to this study: The photoprotective effect of VP extracts was only evaluated using the HaCaT cell model. Future studies will need to investigate the protective effect and its mode of action in other skin cells. In addition, the RNA-seq results need to be validated by pathway inhibition/activation assays. Although the solvents used in this study are considered safe and are commonly used in cosmetic formulations, their specific effects on cells should be further investigated and compared with traditional solvents.

## 5. Conclusions

In conclusion, our results showed that *Viola philippica* polyol extracts have great potential as natural antioxidants and photoprotectants. VP extracts can protect HaCaT cells from UVB-induced damage by enhancing cell viability, reducing intracellular ROS levels, and preventing DNA damage. The protective effects are associated with the regulation of key molecular pathways, including neutrophil extracellular trap formation and the IL-17, HIF-1, and TNF signaling pathways. In addition, VP extracts are safe and biocompatible enough to be used directly as cosmetic ingredients and are therefore expected to be used in cosmetic formulations targeting skin photoaging and oxidative stress.

## Figures and Tables

**Figure 1 antioxidants-14-00884-f001:**
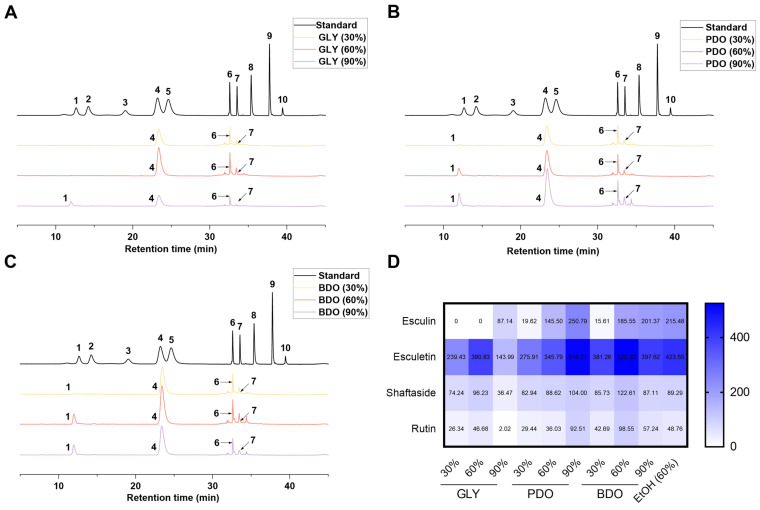
HPLC graph of VP polyol extracts. (**A**) HPLC of VP glycerol extracts; (**B**) HPLC of VP 1,3-propanediol extracts; (**C**) HPLC of VP 1,3-butanediol extracts; (**D**) heatmap of active compound content. 1, esculin; 2, caftaric acid; 3, chlorogenic acid; 4, esculetin; 5, caffeic acid; 6, schaftoside; 7, rutin; 8, chicoric acid; 9, luteolin; 10, naringenin. GLY (glycerol), PDO (1,3-propanediol), BDO (1,3-butanediol), and their concentrations (%).

**Figure 2 antioxidants-14-00884-f002:**
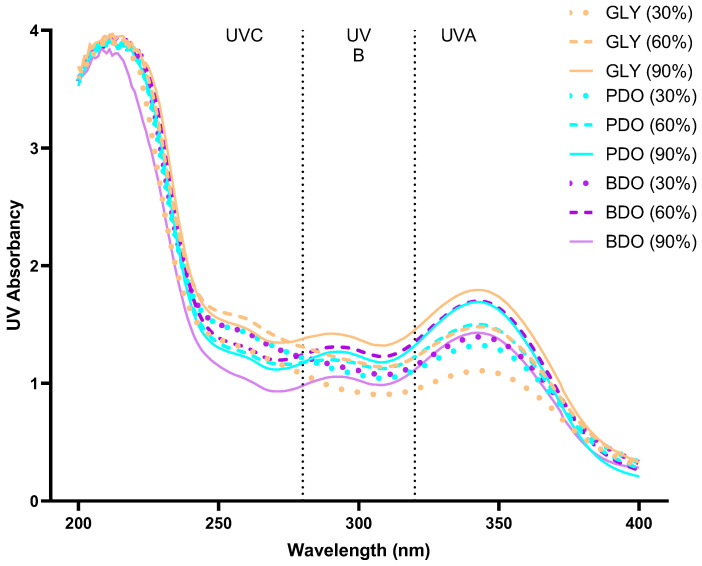
UV wavelength scanning graphs of VP extracts. (A) UV absorption diagram of VP glycerol extracts; (B) UV absorption diagram of VP 1,3-propanediol extracts; (C) UV absorption diagram of VP 1,3-butanediol extracts. GLY (glycerol), PDO (1,3-propanediol), BDO (1,3-butanediol), and their concentrations (%).

**Figure 3 antioxidants-14-00884-f003:**
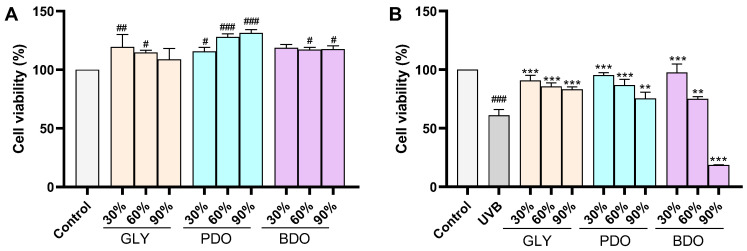
Effect of VP polyol extracts on the activity of UVB-induced HaCaT cells. (**A**) Effect of VP polyol extracts on HaCaT cell viability; (**B**) effect of VP polyol extracts on viability of HaCaT cells irradiated by UVB (0.5 J/cm^2^). Data are presented as the mean ± SD (*n* = 3). # *p* < 0.05, ## *p* < 0.01, and ### *p* < 0.001 vs. control group; ** *p* < 0.01 and *** *p* < 0.001 vs. UVB group (one-way ANOVA, Tukey’s test). GLY (glycerol), PDO (1,3-propanediol), BDO (1,3-butanediol), and their concentrations (%).

**Figure 4 antioxidants-14-00884-f004:**
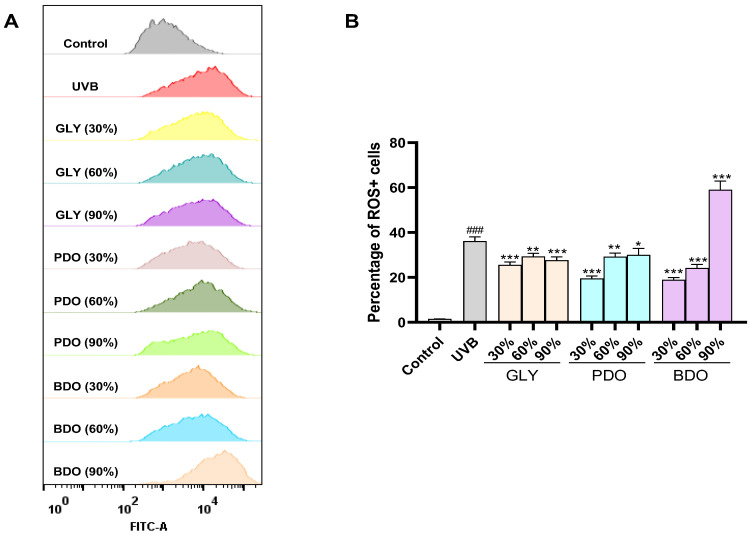
Effect of VP extracts on intracellular ROS levels in HaCaT cells after UVB irradiation. (**A**) Detection of intracellular ROS with VP extracts in UVB-induced HaCaT cells by flow cytometry; (**B**) histogram of intracellular ROS levels. Data are presented as the mean ± SD (*n* = 3). ### *p* < 0.001 vs. control group; * *p* < 0.05, ** *p* < 0.01, and *** *p* < 0.001 vs. UVB group (one-way ANOVA, Tukey’s test). GLY (glycerol), PDO (1,3-propanediol), BDO (1,3-butanediol), and their concentrations (%).

**Figure 5 antioxidants-14-00884-f005:**
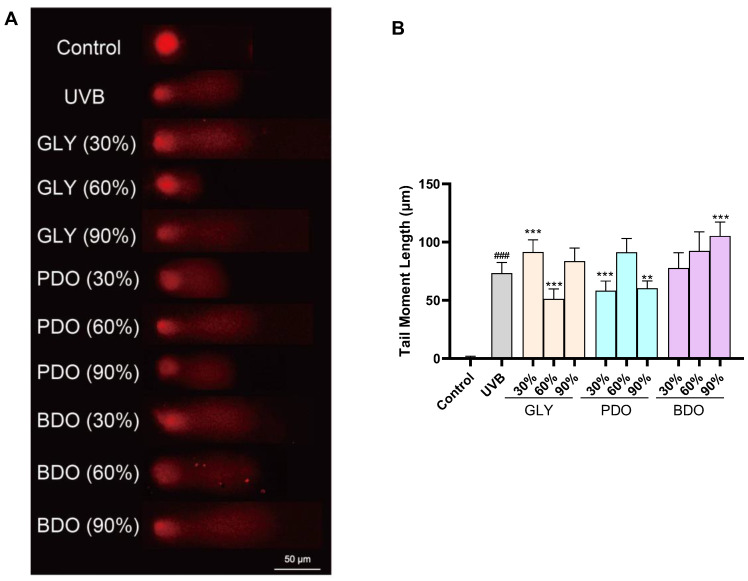
Protective effect of extracts against UVB-induced DNA damage in HaCaT cells. (**A**) Representative graph of DNA damage comets; (**B**) tail moment length statistics. Data are presented as the mean ± SD (*n* = 3). ### *p* < 0.001 vs. control group; ** *p* < 0.01 and *** *p* < 0.001 vs. UVB group (one-way ANOVA, Tukey’s test). GLY (glycerol), PDO (1,3-propanediol), BDO (1,3-butanediol), and their concentrations (%).

**Figure 6 antioxidants-14-00884-f006:**
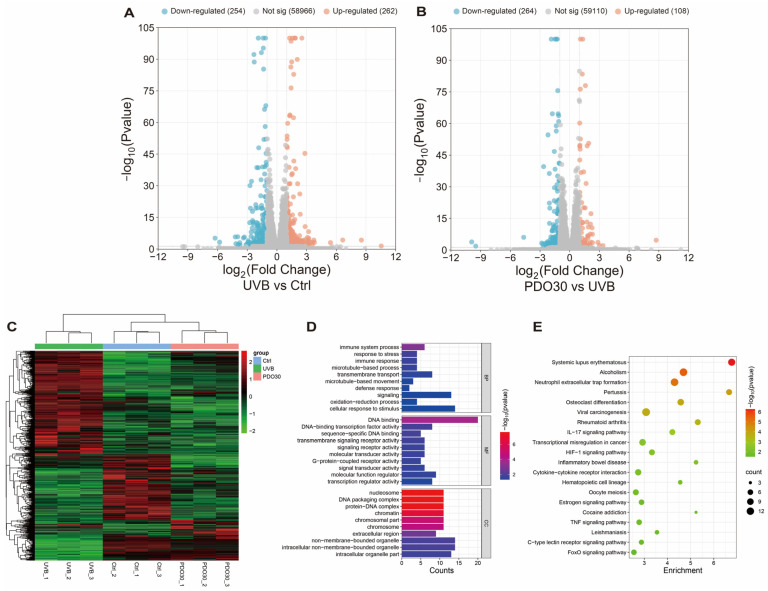
Transcriptomic effects of PDO (30%) on UVB-induced HaCaT cells. (**A**) Volcano plots of mRNA expression in HaCaT cells of UVB and control groups; (**B**) volcano plots of mRNA expression in HaCaT cells of PDO (30%) and UVB groups; (**C**) heatmaps showing mRNA in cells of different groups with hierarchical clustering; (**D**) GO enrichment analysis of the DEGs in the PDO (30%) group vs. UVB group; (**E**) enrichment analysis of the top 20 KEGG pathways in the PDO (30%) group vs. UVB group.

**Table 1 antioxidants-14-00884-t001:** Total phenolic content (TPC) and total flavonoid content (TFC) of VP polyol extracts (mean ± SD, *n* = 3).

VP Extracts	TPC (mg GAEs/g)	TFC (mg REs/g)
EtOH (60%)	63.58 ± 0.05	43.33 ± 0.07
GLY (30%)	57.25 ± 0.19 ***	31.77 ± 0.07 ***
GLY (60%)	64.75 ± 0.45 **	46.68 ± 0.27 ***
GLY (90%)	34.73 ± 0.05 ***	26.68 ± 0.32 ***
PDO (30%)	61.09 ± 0.09 ***	35.12 ± 0.07 ***
PDO (60%)	66.53 ± 0.69 ***	39.71 ± 0.34 ***
PDO (90%)	71.45 ± 0.17 ***	40.62 ± 0.22 ***
BDO (30%)	64.57 ± 0.02 *	34.72 ± 0.29 ***
BDO (60%)	71.28 ± 0.59 ***	40.65 ± 0.19 ***
BDO (90%)	46.23 ± 0.05 ***	32.72 ± 0.14 ***

EtOH (ethanol), GLY (glycerol), PDO (1,3-propanediol), BDO (1,3-butanediol), and their concentrations (%). * *p* < 0.001, ** *p* < 0.001, and *** *p* < 0.001 vs. EtOH (60%).

**Table 2 antioxidants-14-00884-t002:** Summary of the clearance results of DPPH, ABTS, and OH· of different VP extracts (mean ± SD, *n* = 3).

VP Extracts	DPPH Radical IC_50_ (mg/mL)	ABTS Radical IC_50_ (mg/mL)	OH· Radical IC_50_ (mg/mL)
Vc (μM)	20.34 ± 0.1	18.12 ± 0.6	359.15 ± 0.36
EtOH (60%)	0.4 ± 0.06	0.11 ± 0.01	181.6 ± 5.23
GLY (30%)	0.64 ± 0.03	0.11 ± 0.01	1.71 ± 0.17
GLY (60%)	0.49 ± 0.05	0.1 ± 0.01	3.65 ± 0.23
GLY (90%)	1.26 ± 0.07	0.11 ± 0.01	3.62 ± 0.42
PDO (30%)	0.6 ± 0.01	0.12 ± 0.01	1.58 ± 0.01
PDO (60%)	0.7 ± 0.01	0.12 ± 0.01	4.19 ± 1.21
PDO (90%)	0.5 ± 0.02	0.12 ± 0.01	339.45 ± 0.92
BDO (30%)	0.89 ± 0.04	0.12 ± 0.01	1.94 ± 0.16
BDO (60%)	0.56 ± 0.02	0.1 ± 0.01	5.95 ± 0.53
BDO (90%)	0.61 ± 0.03	0.19 ± 0.03	460.95 ± 3.82

EtOH (ethanol), GLY (glycerol), PDO (1,3-propanediol), BDO (1,3-butanediol), and their concentrations (%).

## Data Availability

The original contributions presented in this study are included in the article and Appendix A. Further inquiries can be directed to the corresponding author.

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
