# Peer review of "Antioxidant and Photoprotective Activities of Viola philippica Polyol Extracts"

_antioxidants, 2025, doi:10.3390/antiox14070884_

Round 1
Reviewer 1 Report (New Reviewer)
Dear authors,
The study explores polyol-based extraction (using glycerol, 1,3-propanediol, and 1,3-butanediol) of Viola philippica—a less commonly studied method in medicinal plant research. This could influence future standardization for cosmetic formulations, as polyols are both skin-safe and commonly used in topical products.
The focus on Viola philippica is region-specific (mainly China), and the species is not widely recognized in global pharmacopoeias. The findings may have limited international relevance unless more comparative studies with globally accepted herbs are included. Please highlight the importance of this study. More commnets can be found in the recommandations.
Overall, the manuscript can be improved in terms of language clarity and grammar.
Line 336-341: The exception of PDO 90% suggests a possible loss of antioxidant compounds at high solvent concentrations or reduced extraction efficiency, but could be discussed further in the paper. Same counts for all results.
Please add some relevant existing results related to these finds, e.g. the study of Jeong et al., 2020. “Optimization of polyol solvent systems for herbal extract formulation in cosmetics.”, found that moderate concentrations of polyols (30–60%) generally yield better extraction of phenolics and flavonoids. This supports your result that 90% PDO extract was less effective, likely due to reduced solubility of key antioxidant compounds at high solvent concentrations.
Such comparisons in your discussion section will help situate your research within the broader scientific context, reinforce your conclusions, and highlight the novelty of using Viola philippica and polyol-based extraction for cosmeceutical applications.
Author Response
Comments 1: The study explores polyol-based extraction (using glycerol, 1,3-propanediol, and 1,3-butanediol) of Viola philippica—a less commonly studied method in medicinal plant research. This could influence future standardization for cosmetic formulations, as polyols are both skin-safe and commonly used in topical products. The focus on Viola philippica is region-specific (mainly China), and the species is not widely recognized in global pharmacopoeias. The findings may have limited international relevance unless more comparative studies with globally accepted herbs are included. Please highlight the importance of this study. More commnets can be found in the recommandations. Overall, the manuscript can be improved in terms of language clarity and grammar.
Response 1: Thank you for your insightful comments. We have emphasized in the Introduction (Lines 91-93) that polyol-based extraction represents an innovative, green alternative to conventional organic solvents, citing recent advances. This method aligns with the growing demand for skin-compatible botanical extracts in cosmetics.
Comments 2: Line 336-341: The exception of PDO 90% suggests a possible loss of antioxidant compounds at high solvent concentrations or reduced extraction efficiency, but could be discussed further in the paper. Same counts for all results.
Please add some relevant existing results related to these finds, e.g. the study of Jeong et al., 2020. “Optimization of polyol solvent systems for herbal extract formulation in cosmetics.”, found that moderate concentrations of polyols (30–60%) generally yield better extraction of phenolics and flavonoids. This supports your result that 90% PDO extract was less effective, likely due to reduced solubility of key antioxidant compounds at high solvent concentrations.
Such comparisons in your discussion section will help situate your research within the broader scientific context, reinforce your conclusions, and highlight the novelty of using Viola philippica and polyol-based extraction for cosmeceutical applications.
Response 2: Thank you for your constructive suggestion. We have revised the discussion section to incorporate comparisons with existing literature, which helps to contextualize our findings and reinforce our conclusions.
Reviewer 2 Report (New Reviewer)
The article focuses on the antioxidant and photoprotective properties of Viola philippica extracts—a plant traditionally used in Chinese medicine. The authors investigate the effectiveness of extracts prepared with different polyols in protecting skin cells from damage caused by UVB radiation. This topic is highly relevant in the search for natural active ingredients to be used in modern cosmetic formulations with protective and anti-aging properties.
The introduction contains bibliographic items that are about ten years old and older. The introduction should be based on the latest reports that are available on this topic.
How was the herb powder obtained? What were the climatic conditions in which the plants grew? Were they all collected at the same time of the plant's development? Was the material dried? In what way? Did the method of drying and grinding not affect the content of compounds with antiradical activity?
The extraction was carried out at a high temperature (60C). Such a temperature affects the rearrangement of phenolic compounds. It seems that the extraction temperature should be lower. The results of the content of polyphenolic compounds and, above all, their profile depend on it.
What was done with the extracts obtained? Were they immediately subjected to assays or stored? If so, for how long and under what conditions?
The methodology for determining total polyphenols and total flavonoids requires supplementation. From the discussion of the results, it follows that the authors converted the results to ml of extract? There is no information about this in the methodology. This is incorrect. The results should be converted to dry matter so that they can be compared with each other and with the results of other authors.
If the total polyphenols results by the F-C method were converted to gallic acid, why was this compound not determined by HPLC?
For what purpose were two methods used - HPLC and LCMS? In this case it did not make sense. The compounds were already identified using HPLC.
Was the ABTS cation radical solution diluted 30-fold to obtain a specific absorbance, e.g. 0.7?
How was the antiradical activity calculated and expressed in the DPPH and ABTS methods? Don't the authors think that it should be expressed in universal units, e.g. in Trolox? This allows for a better comparison with other results.
The first table presents the results of the total polyphenol content in the different extracts. Were the results statistically significantly different? Table 2 presents the results of antiradical activity expressed as IC50. This is not mentioned in the methodology. Discussion of the results for determining polyphenols and flavonoids is not sufficient. Moreover, the authors in their work, if they want to draw any conclusions about the effectiveness of the solvents used for extraction, would have to compare the effectiveness of such extraction with extraction using traditional methods, e.g. methanol, acetone, ethanol. Perhaps extraction with plain water would be just as effective. This was not done in the work. The discussion did not cite any results from the work of other authors. Therefore, it is not known whether this extraction was effective. The work also used a high, unusual temperature, which is not used for polyphenolic compounds, therefore their profile may be completely different than in traditional extraction methods.
Notes in the description.
Author Response
Comments 1: The introduction contains bibliographic items that are about ten years old and older. The introduction should be based on the latest reports that are available on this topic.
Response 1: Thank you for your constructive suggestion. We have comprehensively updated the Introduction section with recent literature while maintaining key foundational concepts.
Comments 2: How was the herb powder obtained? What were the climatic conditions in which the plants grew? Were they all collected at the same time of the plant's development? Was the material dried? In what way? Did the method of drying and grinding not affect the content of compounds with antiradical activity?
Response 2: Thank you for your valuable feedback. The Viola philippica herb powder used in our study was purchased from Bozhou, Anhui Province, China. The plants were collected at the same time of their development and were already dried prior to our acquisition. The dried material was then ground into a fine powder using a laboratory mill. We acknowledge that drying and grinding processes may affect the content of bioactive compounds. However, further research is needed to investigate the effects of different drying methods on the antioxidant activity of comfrey extracts.
Comments 3: The extraction was carried out at a high temperature (60C). Such a temperature affects the rearrangement of phenolic compounds. It seems that the extraction temperature should be lower. The results of the content of polyphenolic compounds and, above all, their profile depend on it.
Response 3: Thank you for your concern regarding the extraction temperature and its potential impact on phenolic compounds. We acknowledge that high temperatures can affect the stability and profile of these compounds. In our study, the extraction was performed at 60°C based on the study by Wan et al. (10.1021/acssuschemeng.8b05642), which demonstrated that this temperature yields the highest polyphenolic extraction efficiency when using polyol solvents.
Comments 4: What was done with the extracts obtained? Were they immediately subjected to assays or stored? If so, for how long and under what conditions?
Response 4: We appreciate the reviewer's inquiry regarding extract handling. All assays were performed within three months of storage. The extracts were not subjected to any additional processing steps prior to analysis, and all tests were conducted within this six-month period to maintain consistency and reliability of the results.
Comments 5: The methodology for determining total polyphenols and total flavonoids requires supplementation. From the discussion of the results, it follows that the authors converted the results to ml of extract? There is no information about this in the methodology. This is incorrect. The results should be converted to dry matter so that they can be compared with each other and with the results of other authors.
Response 5: Thank you for your insightful comments and suggestions. We have supplemented the relevant details regarding the methods for determining the total polyphenol and total flavonoid content in 2.3 section. The results are expressed as equivalents of gallic acid or rutin per gram of dry matter.
Comments 6: If the total polyphenols results by the F-C method were converted to gallic acid, why was this compound not determined by HPLC?
Response 6: Thank you for raising this important point. Although gallic acid is commonly used in the Folin-Ciocalteu method for standardizing total phenolic content due to its prevalence in many plant extracts, our HPLC analysis did not detect gallic acid in the Viola philippica extracts. This result is consistent with the research by Thonthula et al. (10.3390/ijms252312770), where gallic acid was not the primary compound detected in extracts of related species such as Viola yedoensis.
Comments 7: For what purpose were two methods used - HPLC and LCMS? In this case it did not make sense. The compounds were already identified using HPLC.
Response 7: Thank you for your valuable feedback. While HPLC can effectively separate the components in a mixture, it cannot directly determine the specific structure of a compound based solely on retention time. In contrast, LC-MS confirms the identity of compounds through molecular weight and fragmentation information, thereby enabling a more comprehensive and accurate analysis of the composition of the extract. Typically, the structure of the target compound is first identified through mass spectrometry, followed by the selection of the corresponding standard for quantitative analysis.
Comments 8: Was the ABTS cation radical solution diluted 30-fold to obtain a specific absorbance, e.g. 0.7?
Response 8: Thank you for your question regarding the preparation of the ABTS cation radical solution. The ABTS solution was diluted 30-fold with distilled water to achieve an absorbance of approximately 0.7 prior to use.
Comments 9: How was the antiradical activity calculated and expressed in the DPPH and ABTS methods? Don't the authors think that it should be expressed in universal units, e.g. in Trolox? This allows for a better comparison with other results.
Response 9: Thank you very much for your valuable comments and suggestions. In our study, we determined the free radical scavenging activity using the DPPH and ABTS methods. The antioxidant activity was expressed in terms of the sample concentration (mg/mL) required to achieve a specific scavenging level (the half maximal inhibitory concentration IC50). This approach is relatively common in related studies and provides a direct reflection of the sample's ability to scavenge free radicals. We appreciate your suggestion to express the results in universal units, such as Trolox equivalents, as this facilitates comparison with other research findings. However, the primary focus of our current study is to compare the antioxidant activities among different samples or treatments within the same experimental system. The use of IC50 or similar metrics based on sample concentration more directly reflects the differences in scavenging capacity among samples. In future research, we will prioritize the use of universal units, such as Trolox equivalents, to express antioxidant activity in order to enhance the comparability of our findings.
Comments 10: The first table presents the results of the total polyphenol content in the different extracts. Were the results statistically significantly different? Table 2 presents the results of antiradical activity expressed as IC50. This is not mentioned in the methodology. Discussion of the results for determining polyphenols and flavonoids is not sufficient. Moreover, the authors in their work, if they want to draw any conclusions about the effectiveness of the solvents used for extraction, would have to compare the effectiveness of such extraction with extraction using traditional methods, e.g. methanol, acetone, ethanol. Perhaps extraction with plain water would be just as effective. This was not done in the work. The discussion did not cite any results from the work of other authors. Therefore, it is not known whether this extraction was effective. The work also used a high, unusual temperature, which is not used for polyphenolic compounds, therefore their profile may be completely different than in traditional extraction methods.
Response 10: Thank you for your constructive feedback. We have performed statistical analysis and significance labeling for Total Phenolic Content (TPC) and Total Flavonoid Content (TFC) in Table 1. Additional experimental details regarding IC50 were supplemented in the Materials and Methods section. Information regarding ethanol extracts has been included in Table 1, Table 2, and Figure S1 to enhance comparability, and a comparative study of polyol solvents versus traditional solvents has been introduced in the discussion section.
Reviewer 3 Report (New Reviewer)
The aim of this study was to evaluate the content of phenolic compounds, antioxidant and photoprotective activity of Viola philippica extracts using polyol solvents. The study presents important results, but some aspects of the study need to be improved. One aspect that stands out is the methodology, where several vital pieces of information for understanding the study are missing and have been listed. The conclusion needs to be improved to truly demonstrate the main findings of the study and its importance to the scientific community.
Abstract
The study aims to evaluate the antioxidant and photoprotective effects of the extract of the traditional plant Viola philippica in Chinese medicine. In the abstract, the authors report that they compared 9 types of extracts with different compounds. "VP was extracted using glycerol (GLY), 1,3-propanediol (PDO), and 1,3-butanediol (BDO) at concentrations of 30%, 60%, and 90% (v/v) to evaluate its antioxidant and UV-protective properties." However, at the end of the abstract, they conclude: "These findings suggest that Viola philippica polyol extracts have promising potential for skin photoprotection and could be utilized as natural antioxidants in cosmetic formulations."
Which of the extracts are the authors referring to by "Viola philippica polyol extracts." All or some in particular?
Introduction
It is well written and covers all the topics necessary to inform the reader about what the work will cover.
Overall, the work is written in well-written and easy-to-understand English. However, some passages could have a more scientific tone. I suggest reviewing this.
Methods
line 122. "and then extracted at a constant temperature of 60°C for 2 h." It is already very well described in the literature that most bioactive compounds are thermolabile. How do the authors ensure that no compounds are lost with an extraction at 60ºC for 2 hours?
lines 120-122. "The whole herb powder (10 g) of VP was mixed with 100 mL of glycerol (GLY), 1,3-pro-panediol (PDO) and 1,3-butanediol (BDO) at different concentrations of solvent systems (30%, 60%, 90% (v/v))," What would these solvent systems be? If you are going to use a powder in liquid, you should describe these concentrations used in detail since you are putting them as v/v.
I think it is interesting that the authors are looking for new solvents for the plant extraction process. However, I was wondering if compared to traditional solvents, these would be more or less efficient in extracting the compounds of interest. Please include this at some point in the work.
lines 211-213. Describe the origin of the cells here. Did they come from ATCC or another cell bank?
The data on concentration and time of exposure of cells to the extracts are scattered throughout the text. I suggest bringing this data and explaining it in detail here in section 2.5.1. Furthermore, it is not clear how long the cells were exposed to UVB radiation, and also whether this exposure of the cells was before or after treatment with the extracts. Let's make this clear here in this session.
Furthermore, it is not clear from the methodologies whether the solvents used were removed from the extracts or whether they were part of them. If they were removed, how were they reconstituted? If they were not removed, could the solvents themselves not cause damage to the cells?
If the solvents were not removed, the treatment should have treated the cells only with these solvents as a control. All of this must be described in the methodology.
Results.
In the results section, you should only describe the results, leaving the explanations and citations for the discussion. Or you could combine the results and discussion, making the text clearer.
Discussion
The discussion is well-founded. However, I still wonder what the effect of the solvents used for extraction would have been on the cells. I believe that this is vital data that should be included in the results. Or if these solvents have already been tested on cells and are safe, provide data from the literature.
Conclusion
I believe that the conclusion can be greatly improved. Concluding that the extracts with these solvents had more polyphenolic compounds and greater antioxidant capacity is very important. Including the main findings and limitations of the work in the conclusion would also be very good.
Author Response
Comments 1: The study aims to evaluate the antioxidant and photoprotective effects of the extract of the traditional plant Viola philippica in Chinese medicine. In the abstract, the authors report that they compared 9 types of extracts with different compounds. "VP was extracted using glycerol (GLY), 1,3-propanediol (PDO), and 1,3-butanediol (BDO) at concentrations of 30%, 60%, and 90% (v/v) to evaluate its antioxidant and UV-protective properties." However, at the end of the abstract, they conclude: "These findings suggest that Viola philippica polyol extracts have promising potential for skin photoprotection and could be utilized as natural antioxidants in cosmetic formulations." Which of the extracts are the authors referring to by "Viola philippica polyol extracts." All or some in particular? It is well written and covers all the topics necessary to inform the reader about what the work will cover. Overall, the work is written in well-written and easy-to-understand English. However, some passages could have a more scientific tone. I suggest reviewing this.
Response 1: Thank you for your valuable feedback. We have revised the abstract to clearly indicate that the "Viola philippica polyol extract" refers to extracts with significant antioxidant activity, such as the 60% GLY, 30% PDO, and 90% PDO extracts.
Comments 2: line 122. "and then extracted at a constant temperature of 60°C for 2 h." It is already very well described in the literature that most bioactive compounds are thermolabile. How do the authors ensure that no compounds are lost with an extraction at 60ºC for 2 hours?
Response 2: We appreciate the reviewer's valid concern regarding extraction temperature. We acknowledge that high temperatures can affect the stability and profile of these compounds. In our study, the extraction was performed at 60°C based on the findings of a previous study (10.1021/acssuschemeng.8b05642), which demonstrated that this temperature yields the highest polyphenolic extraction efficiency when using polyol solvents.
Comments 3: lines 120-122. "The whole herb powder (10 g) of VP was mixed with 100 mL of glycerol (GLY), 1,3-pro-panediol (PDO) and 1,3-butanediol (BDO) at different concentrations of solvent systems (30%, 60%, 90% (v/v))," What would these solvent systems be? If you are going to use a powder in liquid, you should describe these concentrations used in detail since you are putting them as v/v.
Response 3: Thank you very much for your insightful comments and suggestions. We apologize for the oversight and have revised the manuscript to provide a more detailed and precise description of the solvent systems. Specifically, we now explicitly state that the concentrations used (30%, 60%, and 90%) refer to the weight percent (w/w) of each polyol in the solvent system.
Comments 4: I think it is interesting that the authors are looking for new solvents for the plant extraction process. However, I was wondering if compared to traditional solvents, these would be more or less efficient in extracting the compounds of interest. Please include this at some point in the work.
Response 4: Thank you for your interest and suggestion. We have added some research on polyol solvent extracts in lines 428-436, which indicates that polyol solvents either outperform traditional solvents or are comparable to them in extracting target compounds.
Comments 5: lines 211-213. Describe the origin of the cells here. Did they come from ATCC or another cell bank?
Response 5: Thank you for your valuable suggestion. We have revised the manuscript to specify that the HaCaT cells were obtained from the Cell Bank of the Chinese Academy of Sciences (Shanghai, China).
Comments 6: The data on concentration and time of exposure of cells to the extracts are scattered throughout the text. I suggest bringing this data and explaining it in detail here in section 2.5.1. Furthermore, it is not clear how long the cells were exposed to UVB radiation, and also whether this exposure of the cells was before or after treatment with the extracts. Let's make this clear here in this session.
Response 6: Thanks to your suggestion. we have revised section 2.5.1 to clearly present all treatment parameters in one place. We have now specified that:Cells were pretreated with 1% VP extracts for 2 hours before UVB exposure,UVB irradiation (0.5 J/cm²) was performed using a Bio-SUN irradiator, and Cells were analyzed 24 hours post-irradiation.
Comments 7: Furthermore, it is not clear from the methodologies whether the solvents used were removed from the extracts or whether they were part of them. If they were removed, how were they reconstituted? If they were not removed, could the solvents themselves not cause damage to the cells?If the solvents were not removed, the treatment should have treated the Response 7: We appreciate the reviewer's important question regarding solvent handling. In our study, all polyol solvents (GLY, PDO, BDO) remained in the extracts and were not removed, as these are biocompatible solvents commonly used in cosmetic formulations.
Comments 8: In the results section, you should only describe the results, leaving the explanations and citations for the discussion. Or you could combine the results and discussion, making the text clearer.
Response 8: Thank you for your valuable suggestions. we have deleted the contents of the results section that are of a discussion
Comments 9: The discussion is well-founded. However, I still wonder what the effect of the solvents used for extraction would have been on the cells. I believe that this is vital data that should be included in the results. Or if these solvents have already been tested on cells and are safe, provide data from the literature.
Response 9: Thank you for your positive feedback.We have indeed evaluated the impact of the solvents on cell viability and have included these results in the supplementary material (Figure S3). Our experiments demonstrated that even when the 90% (w/w) polyols and diluted 100-fold, they did not exert any significant cytotoxic effects on HaCaT cells. These findings suggest that the solvents used in our extraction process are safe for the cells under the conditions employed in our study.
Comments 10: I believe that the conclusion can be greatly improved. Concluding that the extracts with these solvents had more polyphenolic compounds and greater antioxidant capacity is very important. Including the main findings and limitations of the work in the conclusion would also be very good.
Response 10: Thank you for your valuable suggestion. We have carefully revised the conclusion to better reflect the main findings of our study while also acknowledging its limitations.
Reviewer 4 Report (New Reviewer)
the question of selection 344 nm as detection wavelength must be explained? why the authors choose that specific wavelength? for which compounds is specific?
the lines in figure 3 could be more visible at the current stage its pale and its difficult to clarify which is which!
the research topic is very interesting and the presented data could have great impact on the ordinary life and scientific community. the authors investigate various solvent that could be used for extraction of active compounds of Viola philippica (VP), and their potential use in skin protective cream against UV radiation. the article is well organized and presented in logical manner. the introduction is very informative and describe all key topics necessary for proper understanding of the research. the materials and methods precisely describe experimental conditions and redear could easily repeat the experiments if he want.
the results are presented from general to the specific and properly discussed and explained. the conclusion consolidate results and pointed out the most important one.
the number of figures and tables are satisfactory and give additional value to the article by giving opportunity to redear to make conclusions by itself.
Author Response
Comments 1: the question of selection 344 nm as detection wavelength must be explained? why the authors choose that specific wavelength? for which compounds is specific?
Response 1: We greatly appreciate the valuable feedback you provided. We chose 344 nm because it is the specific wavelength for detecting esculetin, which is a characteristic compound in Viola philippica. This wavelength allows for the effective identification and quantification of esculetin in the extracts.
Comments 2: the lines in figure 3 could be more visible at the current stage its pale and its difficult to clarify which is which!
Response 2: Thank you for your valuable suggestions. We have revised Figure 3 to enhance the visibility of the lines.
Comments 3: the research topic is very interesting and the presented data could have great impact on the ordinary life and scientific community. the authors investigate various solvent that could be used for extraction of active compounds of Viola philippica (VP), and their potential use in skin protective cream against UV radiation. the article is well organized and presented in logical manner. the introduction is very informative and describe all key topics necessary for proper understanding of the research. the materials and methods precisely describe experimental conditions and redear could easily repeat the experiments if he want.
the results are presented from general to the specific and properly discussed and explained. the conclusion consolidate results and pointed out the most important one.
the number of figures and tables are satisfactory and give additional value to the article by giving opportunity to redear to make conclusions by itself.
Response 3: Thank you for your valuable suggestions. We have revised Figure 3 (new Figure 2) to enhance the visibility of the lines.
Round 2
Reviewer 1 Report (New Reviewer)
No more comments.
No more comments.
Author Response
We sincerely thank you for your careful review of this article and for your valuable comments.
Reviewer 2 Report (New Reviewer)
The authors suggest that further research is needed on the effect of the type of drying on the content of bioactive compounds, but such studies have been conducted for years and can be easily found in any knowledge base. Bioactive compounds in each plant behave similarly.
If the authors did not dry the plants, then it is impossible to determine what this process looked like and what degree of maturity the plants were.
If gallic acid was not detected in the extracts, then why was the polyphenol content calculated based on it? This is a methodological and substantive error.
The extracts obtained were tested over a period of 3 months and were not frozen? In that case, there was definitely a change in the configuration of polyphenols and the results obtained are not reliable.
Notes in the description.
Author Response
Comments 1: The authors suggest that further research is needed on the effect of the type of drying on the content of bioactive compounds, but such studies have been conducted for years and can be easily found in any knowledge base. Bioactive compounds in each plant behave similarly.
Response 1: We appreciate your insight on this matter. While it is true that many studies have investigated the effects of drying methods on bioactive compounds, our manuscript focuses on the specific application of polyol solvents for extraction, which is a novel approach in the context of our study.
Comments 2: If the authors did not dry the plants, then it is impossible to determine what this process looked like and what degree of maturity the plants were.
Response 2: Thank you for your insightful comment. In response to your query, we would like to clarify that although we did not perform the drying process ourselves, the dried whole plants of Viola philippica used in our study were purchased from Bozhou, Anhui Province, and were identified by Prof. Wu Mingyi of the Kunming Institute of Botany, Chinese Academy of Sciences. The purchase and identification of the plants were strictly based on the Pharmacopoeia of the People's Republic of China 2020 Edition, which records that Viola philippica is harvested and sun-dried during the spring and autumn seasons each year. Therefore, we can confidently state that the plants used in our study were of the appropriate maturity and quality as specified by the Pharmacopoeia of the People's Republic of China. We understand the importance of ensuring the consistency and quality of plant materials in research, and we believe that adhering to the standards set by the Pharmacopoeia of the People's Republic of China provides a reliable basis for our study. As mentioned above, we have added the relevant information to the main text of the herbal medicine source description. “Quality control of herbal medicines is carried out in accordance with the Pharmacopoeia of the People's Republic of China.”
Comments 3: If gallic acid was not detected in the extracts, then why was the polyphenol content calculated based on it? This is a methodological and substantive error.
Response 3: Thank you for your rigorous review of the methodology presented in this paper. The principle of the Folin-Ciocalteu method involves the reduction of the Folin-Ciocalteu reagent by the phenolic hydroxyl groups (-OH) present in polyphenolic compounds under alkaline conditions, resulting in the formation of a blue molybdenum-tungsten blue complex (with a maximum absorption peak at 765 nm). The degree of color development depends on the total amount of reducing groups provided by all phenolic substances in the sample, rather than a specific single compound. Gallic acid, due to its stable structure, good solubility in water, and high sensitivity in color development, is widely selected as a reference compound. The results are expressed as "gallic acid equivalents (mg GAE/g)", meaning "the reducing capacity of total polyphenols in the extract is equivalent to how many milligrams of gallic acid". This is a relative quantitative unit and does not imply that the extract must contain gallic acid. This method has been applied in thousands of phytochemical literature((e.g., https://doi.org/10.3390/antiox14020192, https://doi.org/10.1016/j.foodchem.2020.127836, https://doi.org/10.3390/antiox7120175).).
Comments 4: The extracts obtained were tested over a period of 3 months and were not frozen? In that case, there was definitely a change in the configuration of polyphenols and the results obtained are not reliable.
Response 4: We appreciate your concern about the potential impact on the polyphenol configuration and the reliability of the results. In our study, the extracts were stored at 4°C in the dark without freezing. To mitigate this impact, we filled the storage containers with N2, aiming to minimize the oxidation and degradation of bioactive compounds, thus preserving their structural integrity and activity during storage. However, I apologize that the original manuscript did not provide detailed information, but these details have been added to the paper.

Reviewer 3 Report (New Reviewer)
All considerations were answered and the text was substantially modified. It is now suitable for publication.
All considerations were answered and the text was substantially modified. It is now suitable for publication.
Author Response
We are deeply grateful for your time and expertise in reviewing our manuscript.
Round 3
Reviewer 2 Report (New Reviewer)
Sufficient corrections have been made.
Sufficient corrections have been made.
This manuscript is a resubmission of an earlier submission. The following is a list of the peer review reports and author responses from that submission.
Round 1
Reviewer 1 Report
Dear authors, the manuscript has merit and is within the scope of the journal.
However, in figures 4 and 5, the authors show a partial reversal of the effects caused by UVB. What is the reason? Would this bring any practical advantage?
I suggest that toxicity tests with VP extracts be performed using the lactate dehydrogenase (LDH) release assay. I suggest treating only with the extracts for at least 24 hours.
I suggest adding information about the extracts. After adding the extracts, do they become soluble in different concentrations? An image of this effect in the supplementary figure could be very enlightening.
Do any of the substances identified in the extract have antioxidant action? This would be interesting to add to the discussion.
A good indicator of toxicity of extracts is the comet assay. In this assay, we can observe a reversal. However, the reversal is partial. Therefore, part of the effect may be caused by the extract itself. What is the result of treatment with only extracts, without exposure to UVB?
No
Author Response
Comments 1: Dear authors, the manuscript has merit and is within the scope of the journal. However, in figures 4 and 5, the authors show a partial reversal of the effects caused by UVB. What is the reason? Would this bring any practical advantage?
Responses 1: Thanks for your kind comment. VP extracts partially reversed UVB-induced DNA damage and ROS production. The possible reasons are as follows: firstly, VP extracts have strong UV-absorbing capacity, they can absorb part of UVB and attenuate UVB-induced DNA strand breaks (Figure 3); secondly, VP extracts attenuate UVB-induced oxidative damage by modulating IL-17, TNF, and HIF-1 signaling pathways (Figure 7). This protective effect is important for daily skin care and sunscreen products to effectively delay the onset of skin photoaging and oxidative stress-related diseases. We've added these descriptions to the discussion section.
Comments 2: I suggest that toxicity test-s with VP extracts be performed using the lactate dehydrogenase (LDH) release assay. I suggest treating only with the extracts for at least 24 hours.
Responses 2: Thanks for your good suggestion. We tested the cytotoxicity of VP extracts after treatment of HaCaT cells for 24 h using the LDH release assay and found no cytotoxicity of these extracts (Figure S1), in agreement with the CCK8 results. We have submitted this result to the Supplementary materials.
Comments 3: I suggest adding information about the extracts. After adding the extracts, do they become soluble in different concentrations? An image of this effect in the supplementary figure could be very enlightening.
Responses 3: Thank you for your valuable suggestions. We have provided images of the extracts in the Supplementary materials (Figure S2), and the extracts can be dissolved in solvents at different concentrations. Additionally, the 30% 1,3-propanediol extract has the lightest color, making it more suitable for use in cosmetic formulations where color requirements are stringent.
Comments 4: Do any of the substances identified in the extract have antioxidant action? This would be interesting to add to the discussion.
Responses 4: Thank you for this insightful suggestion. We have revised the ​Discussion section to explicitly address the antioxidant activity of the compounds identified in the Viola philippica (VP) polyol extracts, including ​esculin, ​esculetin, ​schaftoside, and ​rutin.
Comments 5: A good indicator of toxicity of extracts is the comet assay. In this assay, we can observe a reversal. However, the reversal is partial. Therefore, part of the effect may be caused by the extract itself. What is the result of treatment with only extracts, without exposure to UVB?
Responses 5: Thank you for your insightful comment. To directly address your question, we performed ​comet assays on HaCaT cells treated with 1% VP polyol extracts without UVB exposure. The results showed that no significant DNA damage was observed in cells treated with any of the nine VP extracts compared to untreated controls (Figure S3).
Reviewer 2 Report
- Specify the post hoc test used for statistical analysis. Although the manuscript refers to the use of ANOVA, it does not indicate whether a correction for multiple comparisons (such as Bonferroni or Tukey) was applied.
- In some cases, the standard deviation values are relatively small, which is positive, but the sample size is not reported for each experiment, which is crucial for evaluating the robustness of the data. The number of replicates is given in the Methods section.
- Table 1 shows that the 60% BDO extract has the highest phenolic content, while the 60% GLY extract has the highest flavonoid content. However, there is no in-depth analysis in the discussion as to why these specific extracts have a higher content of these compounds compared to the others. Nor is there any discussion of whether these differences might be related to the polarity of the solvents or the solubility of the compounds.
- In the antioxidant activity assays (DPPH, ABTS, OH-), the use of a positive control (such as ascorbic acid or trolox, etc.) is not mentioned. Please indicate the reason for not including these controls.
- The data show that extraction with different polyalcohols affects the antioxidant capacity. However, the discussion mentions that the extract with 30% 1,3-propanediol (PDO (30%)) had a superior effect in reducing DNA damage, but this is not fully consistent with the antioxidant capacity data (DPPH, ABTS). It should be further discussed whether the observed photoprotection is exclusively mediated by antioxidant activity or whether there are other mechanisms at play.
- The extract with 90% glycerol (GLY 90%) showed the highest absorption in the UV spectrum. However, the same extract had the lowest antioxidant activity in the DPPH assay, suggesting that the photoprotection of this extract may be due to UV absorption rather than antioxidant activity. This difference is not mentioned or explained in the discussion.
1. Include the meaning of GLY, PDO, and BDO in the legends of tables and figures where these abbreviations are used.
Author Response
Comments 1: Specify the post hoc test used for statistical analysis. Although the manuscript refers to the use of ANOVA, it does not indicate whether a correction for multiple comparisons (such as Bonferroni or Tukey) was applied.
Responses 1: Thanks for your precious suggestion. we used Tukey post-hoc tests of one-way ANOVA for multiple comparisons, and this information has been added to the “Materials and Methods” section of the manuscript.
Comments 2: In some cases, the standard deviation values are relatively small, which is positive, but the sample size is not reported for each experiment, which is crucial for evaluating the robustness of the data. The number of replicates is given in the Methods section.
Responses 2: Thank you for your valuable suggestions. We have added the number of replicates used for the test in Methods section.
Comments 3: Table 1 shows that the 60% BDO extract has the highest phenolic content, while the 60% GLY extract has the highest flavonoid content. However, there is no in-depth analysis in the discussion as to why these specific extracts have a higher content of these compounds compared to the others. Nor is there any discussion of whether these differences might be related to the polarity of the solvents or the solubility of the compounds.
Responses 3: We sincerely appreciate your review of the article and the valuable comments provided. In response to your concerns regarding the higher content of phenolic and flavonoid compounds in the 60% BDO extract and 60% GLY extract, we have conducted an in-depth analysis and supplemented the discussion section accordingly.
Comments 4: In the antioxidant activity assays (DPPH, ABTS, OH-), the use of a positive control (such as ascorbic acid or trolox, etc.) is not mentioned. Please indicate the reason for not including these controls.
Responses 4: Thank you very much for your valuable suggestions. We conducted the experiment using ascorbic acid as the positive control, and the data has been added to Table 2.
Comments 5: The data show that extraction with different polyalcohols affects the antioxidant capacity. However, the discussion mentions that the extract with 30% 1,3-propanediol (PDO (30%)) had a superior effect in reducing DNA damage, but this is not fully consistent with the antioxidant capacity data (DPPH, ABTS). It should be further discussed whether the observed photoprotection is exclusively mediated by antioxidant activity or whether there are other mechanisms at play.
Responses 5: We greatly appreciate the valuable feedback you provided. We acknowledge that antioxidant activity is indeed one of the important mechanisms of photoprotection, but photoprotection may also involve other mechanisms. The 30% PDO extract demonstrated strong DNA protection in the comet assay, while its DPPH and ABTS antioxidant activities were not prominent. This suggests that the mechanism of photoprotection may not solely rely on antioxidant activity but also involves other pathways. To further explore this phenomenon, we conducted RNA-seq analysis, which revealed that the 30% PDO extract significantly alleviated UVB-induced cellular damage by modulating the IL-17, HIF-1, and TNF signaling pathways.
Comments 6: The extract with 90% glycerol (GLY 90%) showed the highest absorption in the UV spectrum. However, the same extract had the lowest antioxidant activity in the DPPH assay, suggesting that the photoprotection of this extract may be due to UV absorption rather than antioxidant activity. This difference is not mentioned or explained in the discussion.
Responses 6: We greatly appreciate the valuable feedback you provided. As discussed in Lines 423-431, the GLY 90% extract demonstrated the strongest absorption capacity in the ultraviolet absorption experiment, yet it exhibited the lowest antioxidant activity in the DPPH assay. This phenomenon suggests that the photoprotective effect of this extract may primarily rely on its ultraviolet absorption capability rather than its free radical scavenging ability.
Comments 7: Include the meaning of GLY, PDO, and BDO in the legends of tables and figures where these abbreviations are used.
Responses 7: We greatly appreciate the valuable feedback you provided. In all tables and charts where the abbreviations GLY, PDO, and BDO are used, we have added the full names of these abbreviations along with their concentration information.
Reviewer 3 Report
The abstract needs revision to clearly present specific, quantitative results rather than general, subjective statements (e.g., avoid phrases like "rich in polyphenols" or "significant antioxidant activity").
The introduction lacks critical information:
a. Provide a clear hypothesis and rationale behind the choice of polyol solvents.
b. Add recent literature on Viola philippica, clearly specifying its primary bioactive components relevant for UV protection.
c. Include a schematic or illustrative figure depicting skin layers and mechanisms of UV-induced damage for clarity.
Materials and methods section requires substantial improvements:
a. Include full details (manufacturer and country of origin) for all chemicals and instrumentation used.
b. Clearly describe the botanical authentication of Viola philippica and state the origin of the plant material.
c. Justify the decision to use whole herb powder versus specific plant parts with reference to previous research or preliminary findings.
d. Clarify and justify the extraction procedure: was it only maceration, is the duration of 2 hours adequate, and how was this duration selected?
HPLC methodology description is incomplete and currently not reproducible; authors must add precise gradient conditions, post-run time, and clearly specify analytical parameters such as LOD, LOQ, precursor ions, fragment ions, collision energies, and fragmentor voltages.
The results section should be revised for clarity and accuracy:
a. Move statements belonging to the discussion section out of results (e.g., lines 199-200).
b. Clearly specify statistical significance (provide exact p-values, e.g., line 202).
c. Improve Table 1 and Table 2 by adding measurement units, abbreviations explanations, and ensure standardized numerical precision.
The discussion must be significantly expanded and deepened:
a. Provide a clear rationale for choosing polyol solvents already in the introduction and expand on the advantages and disadvantages identified through experimental results.
b. Contextualize and compare your results comprehensively with existing literature.
Clarify explicitly how Viola philippica’s identity was confirmed (e.g., through consultation with a botanist or providing a voucher specimen number).
Provide more comprehensive details regarding the extraction method (maceration conditions, temperature control, stirring, etc.) and explicitly justify why the extraction time of 2 hours was selected.
ABTS and DPPH assays lack procedural clarity; briefly describe each step clearly, especially noting any specific modifications made compared to the originally cited methods.
In the HPLC results section (Section 3.2.):
a. Clearly state which polyphenols were quantified and present quantitative data in a supplementary table.
b. Explain explicitly why certain polyphenols, for which standards were purchased, were not detected in your extracts.
c. Clearly state if the listed contents (lines 217-218) refer to the sum or individual concentrations of the four quantified polyphenols.
Measurement units must be consistent and clearly expressed, especially in Tables 1 and 2. Concentrations of antioxidant assays (DPPH, ABTS) must be reported explicitly as polyphenol equivalents.
Improve Figure 6 by increasing resolution, enhancing visibility, and increasing font sizes to ensure readability and clarity.
Author Response
Comments 1: The abstract needs revision to clearly present specific, quantitative results rather than general, subjective statements (e.g., avoid phrases like "rich in polyphenols" or "significant antioxidant activity").
Responses 1: We sincerely appreciate the reviewer's insightful suggestions regarding the abstract. As recommended, we have comprehensively revised the abstract.
Comments 2: The introduction lacks critical information:
- Provide a clear hypothesis and rationale behind the choice of polyol solvents.
- Add recent literature on Viola philippica, clearly specifying its primary bioactive components relevant for UV protection.
- Include a schematic or illustrative figure depicting skin layers and mechanisms of UV-induced damage for clarity.
Responses 2: We sincerely appreciate the reviewer’s constructive feedback, which has significantly strengthened the clarity and scientific rigor of our manuscript. Below, we address each point raised and detail the revisions made to address these concerns:
- Hypothesis and Rationale for Polyol Solvents (Lines 89–95).
- Recent Literature on Viola philippica’s Bioactive Components (Lines 67–78).
- Schematic of Skin Layers and UV Damage Mechanisms (New Figure 1).
Comments 3: Materials and methods section requires substantial improvements:
- Include full details (manufacturer and country of origin) for all chemicals and instrumentation used.
- Clearly describe the botanical authentication of Viola philippica and state the origin of the plant material.
- Justify the decision to use whole herb powder versus specific plant parts with reference to previous research or preliminary findings.
- Clarify and justify the extraction procedure: was it only maceration, is the duration of 2 hours adequate, and how was this duration selected?
Responses 3: Thank you very much for your valuable comments. We have made changes and additions to the materials and methods:
- We have added complete details of all chemicals and instruments.
- The Viola philippica (VP) was purchased in Bozhou, Anhui Province, and identified by Prof. Wu Mingyi of the Kunming Institute of Botany, Chinese Academy of Sciences.
- The choice of using whole herb powder rather than specific plant parts was mainly since whole herb extracts are usually superior to extracts from single plant parts in terms of biological activity.
- The extraction process was carried out using a simple maceration method, which is suitable for polyol solvents because the high viscosity and low volatility of polyols allow them to maintain a stable extraction efficiency over a long period of time.
Comments 4: HPLC methodology description is incomplete and currently not reproducible; authors must add precise gradient conditions, post-run time, and clearly specify analytical parameters such as LOD, LOQ, precursor ions, fragment ions, collision energies, and fragmentor voltages.
Responses 4: We thank the reviewer for pointing out the incompleteness of the HPLC methodology description and for providing specific suggestions for improvement. In response to the reviewer's comments, we have revised the HPLC methodology section to ensure it is complete and reproducible.
Comments 5: The results section should be revised for clarity and accuracy:
- Move statements belonging to the discussion section out of results (e.g., lines 199-200).
- Clearly specify statistical significance (provide exact p-values, e.g., line 202).
- Improve Table 1 and Table 2 by adding measurement units, abbreviations explanations, and ensure standardized numerical precision.
Responses 5: Thank you for your valuable suggestions. In response to your suggestion about the revision of the results section, we have adjusted the article accordingly:
- we have deleted the contents of the results section that are of a discussion nature (e.g., the original rows 199-200).
- we have clarified the statistical significance of the results.
- we have optimized Tables 1 and 2, supplemented them with explanations of units of measurement and abbreviations, and standardized the precision of the values so that all data are now expressed in mg/mL.
Comments 6: The discussion must be significantly expanded and deepened:
- Provide a clear rationale for choosing polyol solvents already in the introduction and expand on the advantages and disadvantages identified through experimental results.
- Contextualize and compare your results comprehensively with existing literature.
Responses 6: Thank you for your valuable comments. We have expanded and deepened the discussion section.
- firstly, the motivation for choosing polyol solvents was clarified in the introduction, emphasizing their low toxicity and biocompatibility, and the advantages and disadvantages of polyol solvents were explored in detail in the context of the experimental results, e.g. the high extraction efficiency of glycerol for flavonoids and the selectivity of 1,3-butanediol for phenolic acid compounds.
- Secondly, we made a comprehensive comparison of the experimental results with the existing literature and found that the free radical scavenging ability of polyol extracts due to hydroalcoholic extracts.
Comments 7: Clarify explicitly how Viola philippica’s identity was confirmed (e.g., through consultation with a botanist or providing a voucher specimen number).
Provide more comprehensive details regarding the extraction method (maceration conditions, temperature control, stirring, etc.) and explicitly justify why the extraction time of 2 hours was selected.
ABTS and DPPH assays lack procedural clarity; briefly describe each step clearly, especially noting any specific modifications made compared to the originally cited methods.
Responses 7: Thank you for your helpful suggestion. The Viola philippica was identified by Prof. Wu Mingyi of the Kunming Institute of Botany, Chinese Academy of Sciences.
The extraction time of 2 hours was selected based on preliminary experiments. We found that 2 hours was sufficient to achieve optimal extraction efficiency for the bioactive compounds, as determined by monitoring the total phenolic and flavonoid content over time.
The details of ABTS, DPPH, and OH assays have been added to the Materials and Methods section (Lines 170-203) of the revised manuscript.
Comments 8: In the HPLC results section (Section 3.2.):
- Clearly state which polyphenols were quantified and present quantitative data in a supplementary table.
- Explain explicitly why certain polyphenols, for which standards were purchased, were not detected in your extracts.
- Clearly state if the listed contents (lines 217-218) refer to the sum or individual concentrations of the four quantified polyphenols.
Responses 8: We greatly appreciate your valuable feedback.
- In section 3.2, we clearly stated that esculin, esculetin, schaftoside and rutin were quantified and Figure 2D details the quantitative data of these polyphenols in different extracts.
- For polyphenols that were purchased as standards but not detected in the extracts, this may be due to the concentration of the polyphenol being below the detection limit or not being extracted efficiently during the extraction process due to polarity differences or chemical stability issues.
- We make it clear in the text that the levels listed refer to the amount of esculetin, not the sum of the other polyphenols or the individual concentrations.
Comments 9: Measurement units must be consistent and clearly expressed, especially in Tables 1 and 2. Concentrations of antioxidant assays (DPPH, ABTS) must be reported explicitly as polyphenol equivalents.
Improve Figure 6 by increasing resolution, enhancing visibility, and increasing font sizes to ensure readability and clarity.
Responses 9: Thank you very much for your valuable comments, we take your suggestions very seriously and have carefully considered and addressed them. We have unified the units in Tables 1 and 2 to mg/mL, ensuring that all data are expressed with clear and consistent units. Additionally, we have increased the resolution and enhanced the visibility of Figure 6 (new Figure 7) to ensure the readability and clarity of the figure.
Round 2
Reviewer 1 Report
Dear authors, some points mentioned above were not clearly explained and interpreted. In Fig. S1, the authors did not clarify several points. 1- How many times was the assay repeated on different days? 2- How many replicates were used? 3- How long was the treatment? 4- If 50% of the cells are dead in your control, how can this assay be considered reliable and of good quality? 5- If 50% of the cells are dead due to necrosis, how can we interpret Figure 4? 6- If 5% of the cells are dead, how can ROS production be low in the untreated control? 7- Although the LDH assay is more related to necrosis and the comet assay to apoptosis, wouldn't the authors expect to see some degree of degradation in the untreated cells in the comet assay (Fig. S3)? Since 50% of the cells are dead without treatment. Based on this LDH essay, I consider that the scenario the authors are working with does not match the results presented.
The description and interpretation of the results is not correct, especially the interpretation of the manuscript based on the LDH release assay.
Reviewer 2 Report
Thank you for your reply.
No one
Reviewer 3 Report
The authors have clearly put significant effort into revising the manuscript in response to my comments. They’ve addressed each of my concerns with attention to detail, improving the clarity and scientific rigor of the paper. The abstract now presents specific, quantitative results, which was one of my main suggestions. They’ve also strengthened the introduction, providing a clearer hypothesis and more recent literature on Viola philippica.
In the methods section, the authors included the missing details on chemicals, reagents, and instrumentation, and better justified their choice of the extraction method. The HPLC methodology is now much more reproducible, with precise details on gradient conditions and analytical parameters, which is great to see.
The results section is clearer, with added statistical information and improved tables that follow a consistent format. The discussion has also been expanded to more thoroughly address the advantages and disadvantages of polyol solvents, and the authors have done a great job contextualizing their findings within the existing literature.
Overall, the revisions have enhanced both the clarity and depth of the manuscript, addressing my concerns effectively. These changes should certainly strengthen the manuscript's potential for acceptance.
No detailed comments